# Extracellular Vesicles as Biological Shuttles for Targeted Therapies

**DOI:** 10.3390/ijms20081848

**Published:** 2019-04-15

**Authors:** Stefania Raimondo, Gianluca Giavaresi, Aurelio Lorico, Riccardo Alessandro

**Affiliations:** 1Department of BioMedicine, Neuroscience and Advanced Diagnostics (Bi.N.D), Section of Biology and Genetics, University of Palermo, 90133 Palermo, Italy; stefania.raimondo@unipa.it; 2IRCCS Istituto Ortopedico Rizzoli, Via di Barbiano, 1/10, 40136 Bologna, Italy; gianluca.giavaresi@ior.it; 3Touro University Nevada College of Medicine, Henderson, NV 89014, USA; 4Mediterranean Institute of Oncology Foundation, 95029 Viagrande, Italy; 5Institute of Biomedicine and Molecular Immunology “A. Monroy”, National Research Council, 90146 Palermo, Italy

**Keywords:** drug delivery, liposomes, extracellular vesicles, target therapies, plant-derived extracellular vesicles

## Abstract

The development of effective nanosystems for drug delivery represents a key challenge for the improvement of most current anticancer therapies. Recent progress in the understanding of structure and function of extracellular vesicles (EVs)—specialized membrane-bound nanocarriers for intercellular communication—suggests that they might also serve as optimal delivery systems of therapeutics. In addition to carrying proteins, lipids, DNA and different forms of RNAs, EVs can be engineered to deliver specific bioactive molecules to target cells. Exploitation of their molecular composition and physical properties, together with improvement in bio-techniques to modify their content are critical issues to target them to specific cells/tissues/organs. Here, we will discuss the current developments in the field of animal and plant-derived EVs toward their potential use for delivery of therapeutic agents in different pathological conditions, with a special focus on cancer.

## 1. Introduction

In the last decade, we have assisted to significant advances in the field of nanomedicine, in particular in the innovative area called theranostic. Theranostic exploits nanostructured systems containing drugs for therapeutic purposes and natural nanoparticles for diagnostic purposes [1,2].

Many pathological conditions, including chronic inflammatory disorders and cancers, often require prolonged drug administration to patients, resulting in high cumulative doses and significant side effects. For most drugs currently in use, therapeutic activity is determined by the attainment of a sufficient concentration in their target cell, tissue or organ. However, the distribution of the therapeutic agent in the organism depends not only on the route of administration but also on its chemical and physical characteristics as well as the anatomical and physiopathological characteristics of the target areas. In general, the drug is distributed more or less extensively at the systemic level with consequent secondary toxic effects.

For all these reasons, several efforts are being made to minimize local and systemic toxicity of pre-existing drugs, widely used in clinical practice, and to increase tumor selectivity. Therefore, nanotechnologies applied to medicine, ranging from the medical use of nanomaterials to the formulation of new drug delivery systems, have a tremendous potential [3].

In this context, researchers, in particular in the oncologic field, are currently focusing on the development of drug delivery systems to target tissues with the aim of (i) reducing toxicity; (ii) circumscribing the biological effect on a certain type of cells; and (iii) preserving their therapeutic activity.

Tumor tissues have peculiar pathophysiological characteristics that can be exploited for transmission and selective release of drugs, such as enhanced permeability and retention that allows the accumulation of nanoparticles [4,5]; moreover, since cancer cells often express or overexpress specific membrane receptors, nanoparticle surface can be modified to optimize targeting properties [6].

A recent review explored the use of engineered exosomes in musculoskeletal disorders, highlighting possible new applications in regenerative medicine for osteoporosis and osteoarthritis, for the regulation of the immune system in inflammatory-dependent bone diseases, as well as in the treatment of tumors such as osteosarcomas, chondrosarcomas or bone metastases, although the application of exosomes in these tumors is still in the early stages [7].

To date, the delivery systems applied in drug targeting are different in terms of composition, structure, and drug release rate. They include polymer-based microparticulate systems, phospholipid-based vesicular systems (liposomes), micelles, multifunctional dendritic polymers, liquid crystals, nanocapsules, and nanospheres [8,9]. In particular, since the advent of liposomes as drug delivery vehicles, several drugs commonly used in the clinics have been encapsulated to reduce their toxicity. Although the use of these delivery systems has shown numerous pre-clinical advantages [10], an important question concerning the accumulation of these particles in certain organs remains to be solved. In fact, depending on the nanoparticles used, their accumulation was observed in organs such as the brain, lung, liver, and kidneys, resulting in local toxicity.

Here, we will discuss briefly the most relevant approaches for drug delivery, highlighting the critical issues that often do not allow their clinical use. We will then focus on a new and promising approach based on extracellular vesicles (EVs), small lipoproteic structures released in the extracellular space by all cell types, under both physiological and pathological conditions. Among EVs, those released by human mesenchymal stem cells and plant-derived EVs may represent the most promising approaches for the development of new drug delivery systems in the near future.

## 2. Old and New Opportunities for Drug Delivery

### 2.1. Liposomes

Among the systems of drug delivery, those lipid-based are the most studied so far [9]. In particular, liposomes have been experimentally used as drug delivery vehicles since 1970 [11,12]. Liposomes are hollow microspheres typically composed of various types of phospholipids organized in fluid bilayers. The interest for liposomes is related to their membrane (consisting of cholesterol and phospholipids), whose structure and composition resemble the plasma membrane of recipient human cells. Currently, several biotechnological companies employ liposomes for numerous fields of application, including antibiotics, anticancer, and gene therapy [13].

While lipophilic drugs are usually encapsulated inside the lipid bilayer, some hydrophilic drugs may be solubilized within the aqueous core of the liposome. Liposome characteristics such as composition and size can be modified according to the different compounds to shuttle or to decrease the rate of liposome degradation and control the release of their content. It is also possible to increase the affinity of the liposomes for a given tissue by modifying their composition. In order to obtain a functionalization of the formulation, particular membrane proteins can be incorporated into the membrane of the liposome. These techniques, as well as the functionalization of the system, typically allow reduced natural degradation with subsequently increased stability.

Several studies have analyzed the systemic use of these lipid structures for the delivery of small therapeutics, including DNA, antisense oligonucleotides, and siRNA [14]. The promising results obtained have opened the possibility to develop liposome-based systems for gene therapy both ex-vivo and in vivo; however, toxic effects together with their rapid clearance represent main disadvantages, requiring further investigations before their clinical use [15].

Thus, liposomal doxorubicin has been largely studied to increase the therapeutic index in both solid and hematologic cancers [16]. In fact, lower rates of myelosuppression, cardiotoxicity, and alopecia have been reported compared with conventional doxorubicin [17]. However, the physical stability of the suspensions has been largely debated since liposomes are subjected to oxidative degradation. Moreover, the chemical activity of the encapsulated drug is frequently lost, based on the percentage of cholesterol, nature of the phospholipids, size and chemical-physical characteristics of the carried compound. Although liposome modifications to reduce drug loss have been developed, side effects have still been frequently observed.

### 2.2. Extracellular Vesicles

Scientific research, in particular in the biomedical field, has long been interested in the mechanisms that regulate the communication among cells, since it is well known that they need to continuously exchange information with the surrounding environment to perform their functions. In the past, however, the studies were focused to understand the direct cell-cell interactions or the modality by which individual molecules, released by cells, affect nearby or even distant cells. What had previously been recognized as a secretion mechanism by which cells eliminate their waste [18] is now considered a very important cell-cell communication mechanism that attracts the interest of researchers from many disciplines, having given rise to a vibrant scientific community, named International Society for Extracellular Vesicles (ISEV).

EVs generally indicate lipoproteic vesicles that are released into the extracellular space [19], i.e., exosomes, ectosomes, apoptotic and necrotic bodies. They differ in their origin, composition, and size, thus representing a very heterogeneous population that mainly depends on the state of the cell of origin. Although specific features have been proposed for each EV subpopulation, the identification of specific markers of each subgroup is still debated and under investigation. In particular, the possibility to distinguish exosomes from ectosomes is still largely debated [20].

Recently, the minimal requirements for studies on EVs (MISEV) have been published with the participation of a large part of the scientific community in the field [21], representing an update of the guidelines for working with EVs.

Exosomes are cup-shaped vesicles of 30–150 nm, released by all cell types, that originate when a multivesicular body (MVB) fuses with the plasma membrane, releasing exosomes in the extracellular space [22,23], while ectosomes are shed directly from the plasma membrane. Their size and composition overlaps with those of exosomes, although ectosomes include larger vesicles, up to 1 μm. However, given the lack of specific markers for each category of vesicles, the scientific community is currently oriented towards distinguishing and indicating EVs based on size (small, medium or large), density (low, medium or high), positivity to certain markers and the state of the cells from which they originate [21].

EVs carry complex cellular messages, mediated by proteins, lipids and nucleic acids, which are selectively packaged inside the vesicles and transported outside the producing cell. Thanks to Web-based resources such as ExoCarta, EVpedia, and Vesiclepedia [24,25], we are today aware of the content of EVs released from multiple species ranging from protozoa to human.

Although the intracellular origin of these vesicles as well as their content are widely described in the literature, the mechanism by which some macromolecules are selectively packaged within the vesicles has not been fully elucidated yet; therefore, it is not surprising that many recent studies are aimed at identifying the mechanism that controls the sorting of specific miRNAs. Santangelo et al. [26] described the RNA binding protein SYNCRIP (synaptotagmin-binding cytoplasmic RNA-interacting protein) as responsible for the hepatocyte miRNA sorting in exosomes by direct binding; specifically, they found that this binding occurs through a short sequence named hEXO motif. Wani et al. [27] reported that miR-2909 is packaged in tumor exosomes by a specific 3′-end post-transcriptional modification of the miRNA. It is expected that further studies of molecule sorting mechanisms will improve the development of strategies for exosomal content modification in order to develop better drug delivery vehicles.

The specific EV content reflects that of the cell of origin and determines the vesicle biological effect. In fact, EVs interact with target cells triggering in them phenotypic changes. For these reasons EVs are now considered as leading actors of intercellular communication, mediating both physiological as well as pathological responses [28,29].

EVs can be internalized by classic endocytic mechanisms but also by fusion with the plasma membrane. Alternatively, EVs can trigger intracellular pathways in target cells by ligand-receptor interactions or EV membrane proteins can be cut off by proteases and the resulting fragments act as ligands for cell surface receptors [18,30,31].

A better understanding of the surface molecules of the EVs as well as the possibility of modifying them is the basis of several studies, which will be discussed later, aimed at developing EVs as vehicles for targeted therapy.

## 3. EVs as Drug Delivery Vehicles: Source, Loading, and Targeting

EVs represent an opportunity for the research community to transform cellular structures into new forms of treatment for various diseases, exploiting what nature already offers: systems that deliver biological messages addressed to other cells of the organism. The goal of researchers is to convert the biological messages into therapeutic ones. To face this challenge, various aspects of EV research must be taken into consideration; these aspects, preceded by a deeper understanding of EV biology as well as the improvement of biotechnology techniques, will be discussed in the following subparagraphs and can be summarized as follows: the choice of EV cellular source, isolation methods, loading, and engineering approaches for efficient drug targeting (Figure 1).

### 3.1. EV Cellular Source for Theranostics Approach

A number of studies have suggested that the ability of EVs to deliver chemotherapeutic compounds differ depending on the producer cell. A multitude of parameters need to be considered in order to choose the most appropriate cellular system to produce EVs for therapeutic purposes. Immunogenicity, yield, horizontal gene transfer and ability to be genetically manipulated for cell targeting are the EV properties that need to consider for the potential clinical applications.

Interestingly, most of the cells incubated with chemotherapeutic compounds or nucleic acids are able to package these molecules into EVs that can be subsequently collected and used for different purposes. For example, dendritic cells (DC) have been used in several experimental settings as EV donor cells due to their low immunogenicity profile. EVs released from Indoleamine 2,3-dioxygenase (IDO)-expressing DCs have been demonstrated to be immunosuppressive and anti-inflammatory, and are able to reverse arthritis in a murine model of collagen-induced disease [32]. Li and coll. reported that statin-induced immature dendritic cells secrete tolerogenic EVs, which are involved in the suppression of immune responses in a rat model of experimental autoimmune myasthenia gravis. Authors speculated that animals treated with statin-EVs showed increased numbers of Foxp3+ cells in the thymus thus maintaining immunologic tolerance [33]. Using a similar model of myasthenia, Yin and coll. showed, in mice, that exosomes from microRNA-146a overexpressing DCs inhibited the progress of the disease by shifting the T helper cell immunophenotype from Th1/Th17 to Th2/Treg in spleen [34]. As discussed in the next section, the group of Alvarez–Erviti was able to engineer dendritic cells to produce exosomes displaying on their surface the neuron-specific RVG (Rabies Virus Glycoprotein) peptide [35]; DCs were loaded with siRNA against β-secretase or α-Synuclein and the released EVs, systemically injected in a transgenic mouse model, were shown to localize in brain regions pathologically affected by Parkinson’s disease and more importantly to significantly decrease the level of endogenous mouse α-Synuclein [36]. Another cell type that has been often used to produce EVs for possible therapeutic intervention is the macrophage. EVs from macrophages have shown a biological effect on disease model even without a specific drug loading. For example, in a model of axonal regeneration, EVs with functional NADPH (Nicotinamide Adenine Dinucleotide Phosphate) oxidase 2 complexes were released from macrophages and transferred into damaged axons where they oxidized and inactivated PTEN (Phosphatase and Tensin Homolog), stimulating Akt signaling and regenerative outgrowth [37]. Macrophages loaded with catalase, or transfected with catalase-encoded plasmid can release EVs containing the enzyme. Administration of these vesicles, through intranasal injection, allowed EVs to traverse the blood–brain barrier and be incorporated in neurons, astrocytes, and brain microvessel endothelial cells. This treatment ameliorated symptoms in a mouse model of PD [38,39]. Authors hypothesized that the encapsulation of catalase into EVs may protect the enzyme from degradation and reduce immunogenicity, thus improving therapeutic efficacy. In a similar model of Parkinson, Zhao and colleagues showed that genetically modified macrophages expressing the glial-derived neurotrophic factor (GDNF) produce EVs containing the growth factor. Treatment of mice with these vesicles slowed the progression of the disease [40].

Mesenchymal stem cells (MSCs) have received great interest as functional sources of EVs for drug delivery in the treatment of several disorders, due to their ability to repair tissues and to their immunomodulatory properties. Most of the recent literature on the capability of MSCs to contribute to tissue repair seems to suggest that their beneficial effects in clinical settings are not due to the replacement of impaired and diseased cells but to the biological activity of their “secretome” and in particular of EVs. In 2010, a seminal paper of Lai and coll. showed that in a myocardial I/R injury, MSC- derived EVs exert strong protective effects on heart tissues [41]. Other groups then attributed the underlying mechanisms to the transfer of 20 S proteasome [42], cardiomyocyte autophagy [43] or more recently to modification of the polarization status of macrophages via shuttling of miR-182 [44]. MSC-derived EVs were also found to participate in brain functional recovery after different types of injury. Systemic administration of EVs derived from MSCs was shown to exert therapeutic effects after brain injury [45,46]. Zhang and coll. showed that intravenous delivery of MSC-derived EVs improves functional recovery and promotes neuroplasticity in young adult male rats subjected to a controlled cortical impact [47].

A major challenge in the choice of cellular sources for EV production is availability and possibility for scaling up when primary cells are used or risk for horizontal gene transfer when EVs are recovered from immortalized or tumor cells. A possible alternative has been recently suggested by the group of Le that demonstrated the feasibility in the use of human red blood cells to produce EVs for RNA therapies. Red blood cells belonging to O group are accessible in large quantities in the blood bank and there is no risk of horizontal gene transfer since erythrocytes do not contain DNA. Authors demonstrated that a large amount of red cells EVs can be isolated and electroporated with antisense oligonucleotides directed to miR-125b-2, or Cas9 mRNA and gRNA targeting the miR125b-2 human locus [48]. In all cases, the engineered EVs were able to inhibit both in vitro and in vivo leukemia and breast cancer cells growth. In the following paragraphs, we will discuss how the EVs can be isolated, loaded with drugs and siRNAs and targeted to recipient cells.

### 3.2. Isolation Methods

The technologies for EV isolation as well as their further development are critical for future applications of EVs as drug delivery vehicles; moreover, approaches that enable large-scale isolation are clearly required. In fact, the methods used for vesicle isolation determines the sample yield, its purity, and its future application.

To date, the choice of methods used for purification of the vesicles is based on the starting material and on the downstream use of isolated vesicles. The current gold standard for vesicles isolated from cell culture media and some biological fluids is the use of differential centrifugations to remove cells and large cell debris, followed by ultracentrifugation to precipitate EVs [49,50]. In particular, the first centrifugation, at low speed, is required to pellet cells and eliminates cell debris; then, the supernatant is centrifuged at increasingly higher speeds to pellet larger vesicles. Finally, the resulting supernatant is ultracentrifuged to pellet small EVs. The ultracentrifugation step is critical and is often repeated twice in order to wash and further purify the EV pellet.

One of the disadvantages of this approach, thinking about the future clinical use of the EVs, is the contamination of the isolated sample by proteins and lipoproteins that are difficult to eliminate. The combination of ultracentrifugation with a density gradient is often used to remove contaminants from the EV preparation.

The need to isolate EVs from samples of reduced volume has led many companies to develop methodologies based on precipitation by polymers. The use of these methods, although very useful for small size samples, is still debated since the presence of polymers often interferes with sample analysis.

The detailed description of the protein component of the EVs has allowed the identification of those that today we consider the EV markers, i.e., the proteins present both on the surface and inside the vesicles, that allow their characterization upon isolation. Likewise, the study of the proteomic profiles of the EVs is useful in order to determine those markers that specifically identify EVs from one cell type compared to others. The recognition of the EVs on the base of their protein component underlies the development of methods of isolation through immunocapture [51,52]; specifically, this approach aims to isolate the EVs by using antibodies directed against surface proteins [53]. This method can also be used in combination with those described above. Although this approach can be applied either to isolate all EVs or to purify certain vesicle subtypes, to date this method is difficult to use in the case of large volume samples required in the clinical setting.

Other EV isolation techniques that have been developed are based on size exclusion chromatography (SEC) [54,55] and ultrafiltration; these two approaches, which can also be combined, are used to purify EVs based on their size. The use of a chromatographic approach for EV isolation has proved beneficial in the elimination of contaminants such as proteins and lipoproteins [56], allowing accurate analysis of the sample. The method has also been standardized for the isolation of EVs from complex samples, such as biological fluids (plasma, serum, and urine), thus supporting the possible clinical use of SEC [57,58]. Lastly, ultrafiltration represents a suitable approach for rapid EV purification from large volume samples. This technique involves the use of membranes with specific pore size [59,60] that can be decided according to the size of the vesicles to be purified, and that, when combined with SEC, allows a high degree of purification [61]. The possibility that part of the EVs is lost due to adherence to the membranes used or that the vesicles may be damaged during the process are factors requiring consideration for EV manufacturing and scale-up purposes.

### 3.3. Drug Loading Methods

The simplest method of drug loading is incubation of hydrophobic drugs with EVs. They can pass through the lipid bilayer and be incorporated, as demonstrated with curcumin [62]. Although improved efficacy compared to the free drug has been demonstrated both in vitro and *in vivo*, e.g., for paclitaxel and withaferin A by Munagala et al. [63], the main drawbacks, that limit clinical development, are the inefficient loading capacity and the fact that this strategy is generally limited to hydrophobic compounds.

Several groups have investigated active methods such as sonication, transfection of donor cells, electroporation, extrusion, and direct chemical conjugation. Kim et al. [64] have employed sonication to load macrophage-derived EVs with paclitaxel, bypassing the P-glycoprotein-mediated efflux of paclitaxel and thus overcoming drug resistance in MDR1^+^ (Multidrug Resistance Protein 1) MDCK (Madin-Darby Canine Kidney) cells. During co-sonication of exosomes and drugs, the mechanical shear force from the sonicator’s probe deforms the lipid bilayer of the EV membrane, allowing drug entry by diffusion. Apparently, the EV membrane deformation does not significantly affect the proteo-lipid content of the EV membrane, the membrane integrity being restored within an hour [64]. We transfected donor cells to generate targeted EVs able to deliver Imatinib or BCR-ABL siRNA to CML cells in order to overcome pharmacological resistance, showing that modified EVs, containing IL3-Lamp2B and loaded with Imatinib, are able to specifically target tumor cells *in vivo*, causing a reduction in tumor size [65]. Furthermore, we found that modified EVs are able to deliver functional BCR-ABL siRNA to Imatinib-resistant CML cells. Another group treated SR4987 mesenchymal stromal donor cells with a low dose of paclitaxel for 24 h. After 48 h culture, the paclitaxel-loaded EVs isolated from the medium showed anti-proliferative activities against human CFPAC-1 pancreatic cells in vitro [66]. Electroporation of EVs in the presence of doxorubicin was employed to deliver the antitumor agent to a mouse tumor both in vitro and in vivo [67]. The authors found that electroporation of EVs were successful in efficient delivery of doxorubicin. The rationale for the electroporation method is described in paragraph 3.4 below. Two different extrusion approaches have been investigated: in the first, EVs are mixed with the drug of interest and loaded into a syringe-based lipid extruder with porous membranes [68]; in the second approach, cells undergo serial extrusion through polycarbonate membrane filters with decreasing pore sizes, yielding vesicles whose size and protein composition resembles that of EVs; the vesicles are subsequently incubated with the drug of choice [69]. An advantage of the latter method is the high yield of EVs produced, while the passive drug loading has the low-efficiency problem discussed above. Direct drug conjugation to the EV membrane has been achieved by click chemistry, i.e., copper-catalyzed azide alkyne cycloaddition that forms a triazole linkage [70]. Thus, conjugation of azide-fluor 545 to EVs chemically modified with alkyne groups was reported by Smyth et al. [71]. Interestingly, this approach did not result in changes in the size of EVs nor in the extent of EVs associated with recipient cells, which are potential advantages with respect to other loading methods.

### 3.4. siRNA Loading Methods

Delivery of therapeutic siRNA is particularly challenging due to its size, which can obstacle passive diffusion, and to its susceptibility to RNAse-mediated degradation. In this regard, the protection offered by the EV membrane and the capacity to accommodate macromolecules is a big advantage for EV-mediated siRNA delivery. Electroporation is considered one of the best loading methods for siRNA. This technique creates small pores in the EV membrane when an electrical field is applied in the presence of a conductive solution, resulting in the formation of temporary pores in the EV membrane that allows siRNA present in the solution to penetrate inside the EV. Electroporation leads to superior loading of siRNA over chemical transfection [72] and has been demonstrated to be successful with in vivo models [35]. Didiot et al. [73] employed simple co-incubation of siRNAs with EVs. EVs loaded with siRNA targeting Htt mRNA mediated a dose-dependent silencing of Htt mRNA and protein in primary cortical neurons, a cell type difficult to transfect. El-Andaloussi et al. described a protocol for efficient EV-mediated delivery of siRNA in vitro and in vivo that highlights the critical step of the process [74]. Recently, an arrestin domain containing protein 1 [ARRDC1]-mediated microvesicles (ARMMs), a type of membrane-shed ectosomes discovered in Qian Lu’s lab, were reported to selectively and efficiently recruit, package and biologically deliver active siRNAs into recipient cells, thus identifying ARMMs as a versatile platform for intracellular delivery of macromolecules [75].

### 3.5. EV Engineering Approaches for Efficient Drug Targeting

One of the challenges of the drug delivery systems is to optimize their targeting properties in order to release the therapeutic compounds only to a specific area of our body, thus decreasing the amounts to be administered and avoiding systemic toxicity [4,5].

Increasing evidence demonstrated that EVs possess such advantage since it is possible to engineer EV-producing cells. Ohno and colleagues [76] transfected the EV donor cell line HEK293 with pDisplay encoding the transmembrane domain of platelet-derived growth factor receptor fused to a peptide that specifically binds to EGFR (Epidermal Growth Factor Receptor). The EVs released from the engineered HEK293 cells were used to efficiently deliver the let-7a miRNA to EGFR^+^ breast cancer cells, causing tumor growth inhibition.

To enhance the display of targeting peptides on EV surface, several groups have developed engineered vectors containing the gene for a well-characterized protein of the exosomal membrane, such as the lysosomal-associated membrane protein 2B (Lamp2b), fused with the targeting peptides. Cells transfected with the engineered vectors released EVs displaying the targeting peptides on their surface [35,65,77,78]. This approach was first used by Alvarez-Erviti [35] to produce EVs displaying the neuron-specific RVG peptide on their surface; through this approach, the author demonstrated the brain-targeting capability of the engineered EVs that delivered stable siRNA after systemic administration in mice. Wang and colleagues [77] used this system to obtain EVs containing the anti-fibrotic miR-29 with the rabies viral glycoprotein peptide to increase the vesicle uptake by the kidney that expresses the acetylcholine receptor.

Through a similar approach, we developed EVs with IL3 on their surface to vehicle the anti-leukemic drug Imatinib and siRNA to leukemic blasts overexpressing IL3-R [65]. More recently, Limoni et al. [78] developed engineered EVs to deliver siRNA to HER2^+^ breast cancer cells.

In addition to engineering EV-producing cells, several groups are working on EV membrane functionalization techniques. Considering the future clinical application of EVs, these approaches become very important since they should lead to modifications to already isolated EVs rather than to changes in the cell of origin.

Recent studies have reported EV functionalization approaches by EV membrane covalent modification. Ja et al. developed EVs with imaging and therapeutics properties by loading them with paramagnetic iron oxide nanoparticles and curcumin [79]; the glioma-targeting capability of the vesicles was increased by adding the neuropilin-1-targeted peptide to the exosome membrane by click chemistry. Other EV membrane functionalization approaches consist of cationic lipid and pH-sensitive fusogenic peptide (GALA) conjugation to improve EV cellular uptake and cytosolic release [80].

Recently, Pi and colleagues [81] demonstrated how, through RNA nanotechnology approach, it is possible to increase the specificity of EVs for target cancer cells. To this aim, they developed cholesterol-conjugated RNA aptamers harboring a specific targeting domain for cancer cells; these RNA nanoparticles were conjugated with EVs by incubation at 37 °C. They showed that the EVs conjugated with RNA nanoparticles harboring EGFR aptamer, loaded with survivin siRNA, were able to inhibit breast cancer growth in mice.

Although all these approaches for the ex vivo modification of EVs are promising, many aspects need to be considered in order to ensure the stability and the integrity of EVs during the functionalization process.

### 3.6. Spathasomes and Possibility to Vehicle Therapeutic Materials Directly into the Nucleus

Our knowledge on intracellular routes and the subcellular fate of EV content upon internalization remains scarce [82]. We have recently described a novel subcellular structure, composed of late endosomes that penetrate into type II nuclear envelope invaginations (NEI), that shuttles EV-associated proteins and nucleic acids to the nucleus of host cells [83]. A tripartite protein complex, named VOR, formed by the ER-localized vesicle-associated membrane protein (VAMP)-associated protein A (VAP-A), the cytoplasmic oxysterol-binding protein (OSBP)-related protein 3 (ORP3) and late endosome-associated small GTPase Rab7, orchestrates the entry and retention of EV-containing late endosomes in NEI and the subsequent nuclear delivery of EV-derived cargo proteins and nucleic acids [83,84]. Since this double-organelle structure often appears by fluorescence microscopy as a sword in its scabbard, we named it spathasome from the Greek/Latin words “*spathi*/*spatha*” for sword. The entry of late endosomes into NEI is selective because Rab5^+^ early endosomes and Golgi apparatus were excluded and mitochondria seemed to remain at the border or entry of the NEI [83]. On the cytoplasmic side, we also observed microtubules that allow the movement of late endosomes therein. Silencing of VAP-A or ORP3 abrogated the association of Rab7^+^ late endosomes with NEI as well as the transport of endocytosed EV-derived components to the nucleoplasm of recipient cells [84]. The biological relevance of this novel nuclear structure as an intermediate compartment involved in nuclear transfer of EV components was further demonstrated by transcriptomic analysis of MSCs incubated with melanoma-derived EVs in the presence or absence of importazole, a molecule that specifically inhibits the function of importin β1 by altering its interaction with Ran-GTP, and hence nuclear import [83]. Although further studies are needed to clarify the relevance of this novel pathway, we can envision the potential of this mechanism to deliver engineered EVs directly into the nuclear compartment of host cells, for example transporting in the nucleus of cancer cells chemotherapeutics that exert their antitumor activity specifically at the nuclear level.

### 3.7. Route of Administration, Biodistribution and Immunological Aspects of Therapeutic EVs

Besides considering all the aspects discussed above, to employ EVs as drug delivery systems other aspects concerning the route of administration, the biodistribution of vesicles and their clearance as well as the immunological response by the host to these EVs must be taken into consideration [85]. In vivo experiments on animal models shed light on the possibility to efficiently inject EVs via several routes: subcutaneous, intravenous, intraperitoneal, oral or intranasal. To date, most studies were performed by injecting EVs intravenously or intraperitoneally to allow vesicles to reach internal organs distant from the site of injection. The intravenous route was preferred to deliver EV-therapeutic cargoes in breast and liver tumor xenografts [67,76] and in cardiac injury models [86,87]. The intraperitoneal administration was described to study the effect of MSC exosomes on an experimental model of bronchopulmonary dysplasia [88] or to increase the bioavailability of natural substance, such as curcumin [62]. Intranasal administration of EVs was largely used for brain targeting in order to cross the blood–brain barrier [89,90,91,92]. In 2011, this route was used to deliver exosome-encapsulated curcumin into the brain [92]. Very recently, Thomi et al. observed that mesenchymal stem cells that were administered intranasally reduced neuroinflammation in a rat perinatal brain injury model [89]. Although oral administration of EVs might be the most appropriate system, few studies have been done until now and mainly focused on the oral administration of milk- or plant-EVs [93,94]. The choice of the route of administration influences the biodistribution of the vesicles that can be monitored in vivo through the conjugation with lipophilic trackers such as PKH, DiD or DiR and the use of bioimaging technologies. The distribution kinetics of vesicles is usually analyzed within 48 h of administration, although in many studies the organs are harvested within 24 h [85]. For example, the intravenous injection of HEK293T-derived vesicles leads to their accumulation in liver, spleen, lungs and kidneys [95], while they accumulate in the gastrointestinal tracts and pancreas upon intraperitoneal administration [96]. Through the same approach it is also possible to determine vesicle clearance, a critical topic to consider for the clinical use of the vesicles. Some studies showed that vesicles are rapidly cleared from the blood circulation after intravenous injection [97,98].

Although one of the advantages of using EVs, instead of synthetic nanoparticles, is their low immunogenicity, a current limit to their clinical application is due to their immunostimulatory or immunosuppressive effects, which derive from the complex molecular content of the vesicles; some studies are therefore focused on optimizing the EV surface to decrease the unwanted reactions [99]. This point is critical for the further clinical application of EVs.

## 4. Plant-Derived Vesicles

In the past, studies on EVs have mainly focused on vesicles of animal cell origin, but in recent years there has been an increased interest in vesicles introduced daily with our diet, including milk [63,93,100] and vegetables [101]. These vesicles have attracted widespread attention for therapeutic applications because of their lack of toxicity, for the possibility of large-scale production, their intrinsic properties as well as the possibility to vehicle other compounds, such as drugs or small RNA molecules.

One of the first evidence of the existence of exosome-like vesicles in plants came in 2009 from Regente and colleagues [102], which observed small vesicles with a diameter of 50–200 nm in sunflower seeds. Later, many studies focused on a deep characterization of the vesicles and of their bioactive content to better understand their intrinsic properties and possible biotechnological applications. Zhang’s group isolated vesicles from many plant species such as grape [103], grapefruit [104], ginger [105], and broccoli [106], showing that from all these plant species it is possible, by ultracentrifugation, to isolate vesicles with a specific proteomic, lipidomic and transcriptomic profile. In addition, these edible plant-derived vesicles were found to have anti-inflammatory properties. Grape exosome-like nanoparticles (GELNs) were taken up by intestinal stem cells triggering cell proliferation; these vesicles were also able to protect mice from dextran sulfate sodium (DSS)-induced colitis [103]. Similarly, grapefruit-derived nanovesicles (GDNs) targeted intestinal macrophages, leading to anti-inflammatory effects [104].

Recently, our research group isolated vesicles from citrus-limon juice, with sizes and cargo attributable to exosome-like nanoparticles; we also showed that these vesicles are able to inhibit tumor cell growth by inducing TRAIL (Tumor Necrosis Factor-Alpha-Related Apoptosis-Inducing Ligand)-mediated cell death without affecting normal cells [107].

The possibility of using edible plant-derived vesicles for the loading of other compounds, of vegetal or synthetic origin, for therapeutic purposes appears very promising. For example, grapefruit vesicles loaded with methotrexate and administered to mice with acute colitis had a greater therapeutic index than methotrexate alone [104]. The same grapefruit vesicles were used for the intranasal administration of a specific microRNA, miR-17, leading to the reduction of brain tumor growth in mice [108].

Even in the case of plant vesicles, although these represent a promising opportunity in the future of drug delivery, especially for the possibility of producing them in large quantities and at low cost, the engineering techniques require improvement.

In Table 1, we have summarized the experimental studies on the use of animal and plant EV for therapeutic purposes.

## 5. Conclusions

The attention of many researchers today is not only aimed at the development of new therapies but also at improving the effectiveness of those already used in the clinics. For this reason, many studies are focused on the development of new drug delivery systems that are able (i) to increase the bioavailability of the therapeutic molecules and (ii) to deliver these directly to the target site.

The discovery that EVs, physiologically released by cells of animal and plant origin, are natural vehicles of cellular messages that are able to interact with other cells, has increased the interest of researchers in the field and studies are now aimed at better investigating if they may represent an opportunity as delivery systems.

Although the results produced so far are encouraging, particularly those with MSC-derived EVs and represent a promising approach in the field of drug delivery, many challenges remain to be addressed. In particular, the low-production scale together with the high-production costs due to the sophisticated technologies required, represent limits to their clinical application. In addition, the regulatory aspects of the clinical use of EVs have not yet been defined. The great interest of the community towards the topic is shown by the fact that during the last annual meeting of the International Society of EVs (ISEV, Barcelona 2018), a special session was held to discuss manufacturing license vs. market authorization, regulatory aspects of EVs to reach the clinic, and EVs as medicinal product.

In our opinion, the in-depth characterization of isolated vesicles and their content, together with the standardization of isolation processes, are the most relevant steps required for the clinical application of these biological shuttles.

## Figures and Tables

**Figure 1 ijms-20-01848-f001:**
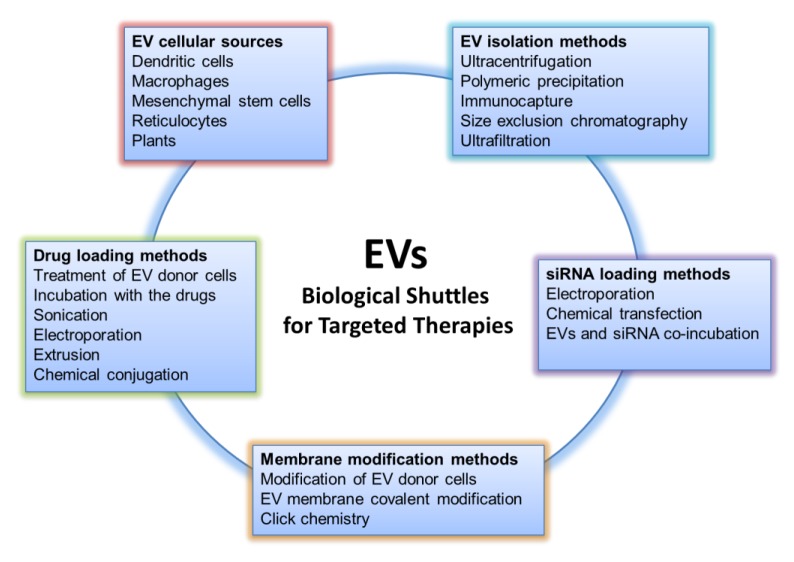
Aspects to be considered for the use of Extracellular Vesicles as drug delivery systems.

**Table 1 ijms-20-01848-t001:** Experimental studies on the use of animal and plant EVs for therapeutic purposes.

EV Source	EV Loading and Engineering Approaches	Functional Effects	References
Dendritic cells (DCs)	Cell transfection Electroporation	DCs were engineered to express the exosomal membrane protein Lamp2b fused with the neuron-specific RVG peptide; EVs were loaded with exogenous siRNA and their administration in mice lead to the downregulation of BACE1, a therapeutic target in Alzheimer’s disease [35]. RVG-exosomes were loaded with α-Synuclein siRNA; their administration in mice lead to the reduction of protein aggregates [36].	[35,36]
Macrophages	Cell transfection Sonication Electroporation	Intranasal administration of EVs containing catalase ameliorated symptoms in a mouse model of PD **38**. Macrophages expressing the glial-derived neurotrophic factor (GDNF) produce EVs containing the growth factor; treatment of mice with these vesicles slowed the progression of the disease [35]. Kim et al. [64] have employed sonication to load macrophage-derived EVs with paclitaxel, bypassing the P-glycoprotein-mediated efflux of paclitaxel and thus overcoming drug resistance in MDR1+ MDCK cells. We developed EVs with IL3 on their surface to vehicle the anti-leukemic drug Imatinib and siRNA to leukemic blasts overexpressing IL3-R [65]. Limoni et al. [78] developed engineered EVs to deliver siRNA to HER2+ breast cancer cells.	[38,40,64,65,78]
MSCs	Incubation of EV-producer cells with the drug	Paclitaxel-loaded EVs isolated from the medium showed anti-proliferative activities against human pancreatic cells in vitro [66].	[66]
Red blood cells	Electroporation	Red cells EVs, electroporated with antisense oligonucleotides directed to miR-125b-2, or Cas9 mRNA were able to inhibit both in vitro and in vivo leukemia and breast cancer cells growth [48].	[48]
Plants	Incubation of plant EVs with the drug or therapeutic nucleic acids	Grapefruit vesicles loaded with methotrexate and administered to mice with acute colitis had a greater therapeutic index than methotrexate alone [104]. Grapefruit vesicles were used for the intranasal administration of miR-17 leading to the reduction of brain tumor growth in mice [108].	[104,108]

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
