# Peer review of "Extracellular Vesicles as Biological Shuttles for Targeted Therapies"

_ijms, 2019, doi:10.3390/ijms20081848_

Round 1

Reviewer 1 Report

The authors submitted their manuscript with the title of Extracellular Vesicles as Biological Shuttles for Targeted Therapies, which is a review. The topic is interesting and has lot of clinical relevance. The authors show older and newer drug delivery possibilities (liposomes and extracellular vesicles).

I suggest to discuss the extracellular vesicles separately, like exosomes, ectosomes, apoptotic bodies, etc. as the use of the term extracellular vesicles could be misleading.

I recommend to create a table showing the types of extracellular vesicles, with their main physical characteristics and applicability as drug delivery system.

I would like to see another table showing the performed studies with different types of extracellular vesicles and the carried drugs (examples)

Immunological aspects of the use of EVs should be discussed (short).

Would be useful to see the perspectives on this field.

Authors contribution part is not filled out, please make it.

Author Response

April 11, 2019

Editorial Office

Dear Editor,

 We are submitting a revised version of the manuscript entitled "Extracellular Vesicles as Biological Shuttles for Targeted Therapies" for publication in the International Journal of Molecular Sciences, for the special issue "Focus on Exosome-Based Cell-Cell Communication in Health and Disease", along with point-by-point answers to the Reviewers.

We would like to thank the Reviewers for their relevant and valuable criticisms and suggestions that definitely helped to improve the quality of the revised version of the manuscript.

Reviewer #1: The authors submitted their manuscript with the title of Extracellular Vesicles as Biological Shuttles for Targeted Therapies, which is a review. The topic is interesting and has lot of clinical relevance. The authors show older and newer drug delivery possibilities (liposomes and extracellular vesicles).

- I suggest to discuss the extracellular vesicles separately, like exosomes, ectosomes, apoptotic bodies, etc. as the use of the term extracellular vesicles could be misleading.

- I recommend to create a table showing the types of extracellular vesicles, with their main physical characteristics and applicability as drug delivery system.

We thank the Reviewer for giving us the opportunity to discuss this point; however, we respectfully disagree with the Reviewer on these suggestions. Although many studies refer to extracellular vesicles as exosomes, ectosomes or microvesicles, at this time no difference in size or content allows a clear distinction among sub-classes, the only difference being biogenetic. This point is supported by the MISEV 2018 recommendation recently published by the EV scientific community (Théry, C.; Witwer, K.W.; Aikawa, E.; Alcaraz, M.J.; Anderson, J.D.; Andriantsitohaina, R.; Antoniou, A.; Arab, T.; Archer, F.; Atkin-Smith, G.K., et al. Minimal information for studies of extracellular vesicles 2018 (misev2018): A position statement of the international society for extracellular vesicles and update of the misev2014 guidelines. Journal of Extracellular Vesicles 2018). Therefore, in line with the MISEV 2018 guidelines, in this review we refer in general to EVs, without specifying the subclasses. We have now clarified this point in section 2.2, line 129.

- I would like to see another table showing the performed studies with different types of extracellular vesicles and the carried drugs (examples)

We agree with the Reviewer that the addition of a table would be useful to summarize the content of this review. Therefore, we have added Table 1 after section 4.

- Immunological aspects of the use of EVs should be discussed (short).

We agree with the Reviewer that the discussion of this important topic was missing in the manuscript. We have added a section, paragraph 3.7 "Route of administration, biodistribution and immunological aspects of therapeutic EVs".

- Would be useful to see the perspectives on this field.

We have added in the conclusion section a paragraph on future perspectives in this field as well as current challenges.

- Authors contribution part is not filled out, please make it.

We have now filled out the authors’ contribution section, as recommended.

Reviewer 2 Report

In the present Review, the authors discuss the potential use of extracellular vesicles (EVs) as a way to efficiently deliver therapeutic agents for the treatment of different diseases. They summarized recent works, showing the latest discoveries in the management of the transport and delivery of the EVs cargo.  They report on the latest discoveries that display potential in cancer treatment. 

This field is still emerging and needs more in-depth studies. The authors are aware of this limitation and they mention it in their manuscript.

Few very minor comments below need to be addressed to improve this Review.

Comments:

·   Regarding the immune surveillance against EVs, their biodistribution and their clearancefollowing systemic administration, the authors are invited to discuss these topics in a separate section with references.

·   In Page 2, Line 3: ‘’  on a certain type of cells; (iii) preserving their therapeutic activity.’’Should add ‘’and’’ to read ‘’  on a certain type of cells; and (iii) preserving their therapeutic activity.’’

·   In Figure 1. Correct to ‘’Chromatography’’.

Author Response

April 11, 2019

Editorial Office

Dear Editor,

 We are submitting a revised version of the manuscript entitled "Extracellular Vesicles as Biological Shuttles for Targeted Therapies" for publication in the International Journal of Molecular Sciences, for the special issue "Focus on Exosome-Based Cell-Cell Communication in Health and Disease", along with point-by-point answers to the Reviewers.

We would like to thank the Reviewers for their relevant and valuable criticisms and suggestions that definitely helped to improve the quality of the revised version of the manuscript.

Reviewer #2: In the present Review, the authors discuss the potential use of extracellular vesicles (EVs) as a way to efficiently deliver therapeutic agents for the treatment of different diseases. They summarized recent works, showing the latest discoveries in the management of the transport and delivery of the EVs cargo.  They report on the latest discoveries that display potential in cancer treatment. This field is still emerging and needs more in-depth studies. The authors are aware of this limitation and they mention it in their manuscript. Few very minor comments below need to be addressed to improve this Review.

- Regarding the immune surveillance against EVs, their biodistribution and their clearance following systemic administration, the authors are invited to discuss these topics in a separate section with references.

We agree with the Reviewer that the discussion of these important topics was missing in the manuscript and we have therefore added a section, paragraph 3.7 "Route of administration, biodistribution and immunological aspects of therapeutic EVs".

- In Page 2, Line 3: ‘’ on a certain type of cells; (iii) preserving their therapeutic activity.’’ Should add ‘’and’’ to read ‘’  on a certain type of cells; and (iii) preserving their therapeutic activity.’’

We have corrected the sentence accordingly.

-In Figure 1. Correct to ‘’Chromatography’’.

We have corrected the figure as requested.

Reviewer 3 Report

Thank you for this beautiful paper. Really well written.

few comments to improve it.

-I don't think  theranostic  should be explained. "(fusion of the terms therapeutic and diagnostic)" can be removed.

-In the first paragraph, please mention brain and lung toxicity because some nanoparticles have been reported to be accumulated in brain and lung and some have not. Although spleen, kidney and liver are common. For example" Karamched, Saketh R., et al. "Site-specific chelation therapy with EDTA-loaded albumin nanoparticles reverses arterial calcification in a rat model of chronic kidney disease." Scientific reports 9.1 (2019): 2629" never reported lung accumulation.

-I think the gold standard method of isolation of EVs must be explained rather than just being referred in line 223.

Author Response

April 11, 2019

Editorial Office

Dear Editor,

 We are submitting a revised version of the manuscript entitled "Extracellular Vesicles as Biological Shuttles for Targeted Therapies" for publication in the International Journal of Molecular Sciences, for the special issue "Focus on Exosome-Based Cell-Cell Communication in Health and Disease", along with point-by-point answers to the Reviewers.

We would like to thank the Reviewers for their relevant and valuable criticisms and suggestions that definitely helped to improve the quality of the revised version of the manuscript.

Reviewer #3: Thank you for this beautiful paper. Really well written. Few comments to improve it.

-I don't think  theranostic  should be explained. "(fusion of the terms therapeutic and diagnostic)" can be removed.

Thank you for the positive comment. We have removed the explanation, as recommended.

-In the first paragraph, please mention brain and lung toxicity because some nanoparticles have been reported to be accumulated in brain and lung and some have not. Although spleen, kidney and liver are common. For example" Karamched, Saketh R., et al. "Site-specific chelation therapy with EDTA-loaded albumin nanoparticles reverses arterial calcification in a rat model of chronic kidney disease." Scientific reports 9.1 (2019): 2629" never reported lung accumulation.

As suggested, we now discuss this point at the end of the first paragraph.

-I think the gold standard method of isolation of EVs must be explained rather than just being referred in line 223.

In the new version of the paper, we have improved this section by adding a deeper description of the EV isolation method by ultracentrifugation.